# A machine learning contest enhances automated freezing of gait detection and reveals time-of-day effects

Amit Salomon[1], Eran Gazit[1], Pieter Ginis[2], Baurzhan Urazalinov, Hirokazu Takoi, Taiki Yamaguchi, Shuhei Goda, David Lander, Julien Lacombe, Aditya Kumar Sinha, Alice Nieuwboer[2], Leslie C. Kirsch[3], Ryan Holbrook[4], Brad Manor[5,6,7] & Jeffrey M. Hausdorff[1,8,9,10] ✉

Freezing of gait (FOG) is a debilitating problem that markedly impairs the mobility and independence of 38-65% of people with Parkinson's disease. During a FOG episode, patients report that their feet are suddenly and inexplicably "glued" to the floor. The lack of a widely applicable, objective FOG detection method obstructs research and treatment. To address this problem, we organized a 3-month machine-learning contest, inviting experts from around the world to develop wearable sensor-based FOG detection algorithms. 1,379 teams from 83 countries submitted 24,862 solutions. The winning solutions demonstrated high accuracy, high specificity, and good precision in FOG detection, with strong correlations to gold-standard references. When applied to continuous 24/7 data, the solutions revealed previously unobserved patterns in daily living FOG occurrences. This successful endeavor underscores the potential of machine learning contests to rapidly engage AI experts in addressing critical medical challenges and provides a promising means for objective FOG quantification.

Freezing of gait (FOG) is a perplexing, disabling symptom that affects 38–65% of patients with Parkinson's disease (PD)[1]. FOG manifests as sudden, unpredictable episodes of being unable to initiate or continue walking despite the intention to do so. Although the neurological mechanism(s) are unclear[2–4], it is clear that FOG profoundly impacts quality of life and is associated with a high fall risk[5–7] and non-motor symptoms such as depression, anxiety, and cognitive impairment[8–10].

To achieve a better understanding of FOG and advance treatment efforts, accurate, objective assessment methods are essential[4,11]. However, the mysterious nature of FOG makes measurement challenging. FOG has multiple physical manifestations, including trembling in place, short shuffling steps, and complete akinesia[12]. It can be triggered under a variety of circumstances, typically when trying to initiate gait (defined as "start hesitation"), when attempting to make a turn, or even during forward walking[2]. Moreover, FOG often disappears when people with FOG walk in front of clinicians[2,13]. This variability in motor manifestation and context makes identifying and quantifying FOG particularly difficult.

Current methods, such as self-report questionnaires like the New Freezing of Gait Questionnaire (NFOG-Q) and visual observation by clinicians, suffer from reliability and subjectivity issues[14]. This leads to huge, almost 2-fold, discrepancies in the estimate of the prevalence of

[1]Center for the Study of Movement, Cognition and Mobility, Neurological Institute, Tel Aviv Medical Center, Tel Aviv, Israel. [2]KU Leuven, Department of Rehabilitation Science, Neuromotor Rehabilitation Research Group (eNRGy), Leuven, Belgium. [3]Michael J. Fox Foundation for Parkinson's Research, New York, NY, USA. [4]Kaggle, San Francisco, CA, USA. [5]Hinda and Arthur Marcus Institute for Aging Research at Hebrew SeniorLife, Boston, MA, USA. [6]Department of Medicine, Beth Israel Deaconess Medical Center, Boston, MA, USA. [7]Harvard Medical School, MA Boston, USA. [8]Sagol School of Neuroscience, Tel Aviv University, Tel Aviv, Israel. [9]Department of Physical Therapy, Faculty of Medical & Health Sciences, Tel Aviv University, Tel Aviv, Israel. [10]Department of Orthopedic Surgery and Rush Alzheimer's Disease Center, Rush University Medical Center, Chicago, IL, USA. ✉e-mail: jhausdor@tlvmc.gov.il

FOG[1]. FOG-provoking stress tests have been used to obtain objective measures of FOG-severity[15,16]. However, they have limitations in reflecting daily FOG occurrence and severity[17–19]. In addition, quantifying the percent time frozen (%TF) during these stress tests, an emerging standard, requires the laborious, time-consuming, offline, frame-by-frame review of videos by two experts[4,20,21]. Furthermore, both stress tests and self-reported outcomes have limitations in reflecting daily FOG occurrence and severity[17–19]. Thus, despite over 30 years of study[22], it is still unknown, for example, if and how FOG frequency changes throughout the day. Hence, accurately capturing the daily-living frequency and variation of FOG on an hour-by-hour basis over multiple days is a key to understanding the impact of medication, daily activity, and fatigue on FOG and eventually advancing effective FOG management.

The rising popularity of applying wearable devices to monitor gait and mobility[23–25], along with significant advances in data science, have led to the pursuit of automatic FOG detection based on the combination of inertial sensors and machine-learning approaches[25,26]. Automatic detection could save the time and the cost of expert, post-hoc review of videos, empower wide-spread testing, improve real-time interventions[13,27], and enable unsupervised, continuous monitoring of FOG during daily living[4,28], an approach that promises ecological validity and relevance to the patient. Since disease manifestations differ in supervised and unsupervised assessments in PD[29–31], since FOG is prone to the white-coat syndrome[2,13], and since factors that may predispose to FOG, like anxiety, depression, and the environment, may fluctuate throughout the day[8–10], testing in the daily living setting is especially valuable for the evaluation of FOG among patients with PD.

Recent studies have applied automatic detection methods to inertial recordings of body movement during FOG-provoking tests and scripted activities of daily-living (ADL)[18,32–42], based on a variety of datasets, detection approaches, and sensor positions. While the results are promising, detection performance has primarily been validated in small samples, or with minimal reporting of metrics like precision and recall (i.e., sensitivity). Furthermore, many commonly used classification metrics like accuracy and ROC curves are less ideal on their own for assessing FOG detection, a naturally imbalanced problem (from the machine-learning perspective) with a low representation of the positive class (i.e., FOG). Some studies utilized data from multiple sensors placed on the body at once[19,32,35], setups that could reduce adaptability and compliance in practice. Moreover, only a few studies have attempted to automatically detect FOG episodes using data captured from unsupervised, habitual daily living activity[17–19], and they have produced only limited and inconsistent results to date.

Multiple analytic approaches have been employed to automatically detect FOG using wearable sensors. Initially, simple threshold-based approaches were introduced[26,43]. Traditional machine learning (ML) algorithms such as support vector machine[32,33,39,42], random forest[33,35,39,40,44,45], and others[33,44,45] followed, offering improved detection abilities. More recently, deep learning models, including convolutional neural networks (CNNs)[34,36–39,41,46], recurrent neural networks (RNNs)[47], and combinations thereof[39,40], have gained popularity due to their theoretically enhanced performance, along with transformers[40] and autoencoders[48]. Still, many challenges persist, including generalizability, overfitting, and limited precision of the tested models.

To expedite the development of a reliable, cost-effective, and widely applicable automatic FOG detection method utilizing only a single inertial measurement unit (a relatively simple configuration with potential widespread use), we organized a FOG detection challenge for the machine learning community using an open-access platform and offering a $100,000 prize, split among the top 5 finishers. The primary aim was to foster the creation and testing of advanced, automated machine learning algorithms to reduce the need for time-consuming video analysis of FOG-provoking tests. More specifically, we aimed to

achieve excellent accuracy and good results regarding precision and reliability. Additionally, we sought to introduce machine-learning experts to PD and FOG. Finally, we explored the application of the winning models to unsupervised, 24/7 real-world FOG data collected over seven days to investigate, on a preliminary basis, the daily patterns (across days and within days) of FOG occurrence in different PD subgroups (patients with reported FOG [freezers] and those without [non-freezers]). Here, we present the results of this competition, discuss the insights gained, and explore its implications.

## Results

### The Parkinson's freezing of gait prediction challenge

Teams participating in this contest developed a machine-learning algorithm for the automated detection and classification of three types of FOG episodes (start hesitation, FOG during turns, during walking), based on 3D acceleration data collected from a single, lower-back sensor and videos that were manually reviewed by experts and fully labeled. 10,133 registrations were made and 1379 teams from 83 countries submitted 24,862 submissions over a competition period of three months.

In over 90 h of recorded data, almost 5000 FOG episodes were identified. Submissions were automatically ranked during the competition based on performance on a randomly selected public test set with 26 patients and 945 FOG episodes. Performance scores on the public test set were revealed to the teams during the competition, but labels of FOG and FOG-class were not. The final ranking was based on a hidden, randomly selected, private test set of 14 patients with 391 validated FOG episodes. In both cases, mean average precision (averaged across three FOG classes) was the scoring metric. This evaluation method is less prone to class imbalance distortions and considers false positive predictions. The top five models are described in the supplementary information. The private mean average precision of the top five ranking models was 0.514, 0.451, 0.436, 0.417, and 0.390, respectively. More details on the competition, scoring, rules, and the final leaderboard can be found online[49,50].

Figure 1 shows the leaderboard scores on both the public and private test sets. After the top five scores on the private leaderboard, there is a drop-off followed by a plateau, where most teams are ranked. This plateau is followed by a further drop in performance starting from approximately the 950th ranked entry. This pattern is similar to the Public leaderboard.

### Post-competition analyses

Precision-recall and receiver operating characteristic (ROC) curves for each of the top five models are shown in Fig. 2. The area under the ROC curves (AUC) was generally high (above 0.9) when looking over all FOG cases (FOG detection regardless of class). Precision-recall curves provide additional information to the ROC curves in cases of class imbalance. The precision-recall curves in these cases are visibly different between classes, showing better performance for the most common class (i.e., FOG during turns). There appears to be a trade-off across the models between successfully detecting Walking FOG events and Start Hesitation FOG events. As most models performed quite well on the Turn FOG class, the 1st place model's primary advantage over the others seems to be the ability to balance best between the other two classes of FOG. The Walking class (which was much less common than the Turn FOG class but more common than start hesitation) has worse ROC curves compared to the other classes in all five models.

F1 scores, accuracy, precision, recall, and specificity are detailed in Table 1. The models show good accuracy (0.88–0.92) when looking at all FOG classes together; each model otherwise showed stronger performance in only one of Walking or Start Hesitation. Specificity was also high (>0.9) for all FOG; recall was relatively low in comparison, but still good (0.72–0.79). Precision ranged between 0.74 and 0.84, with

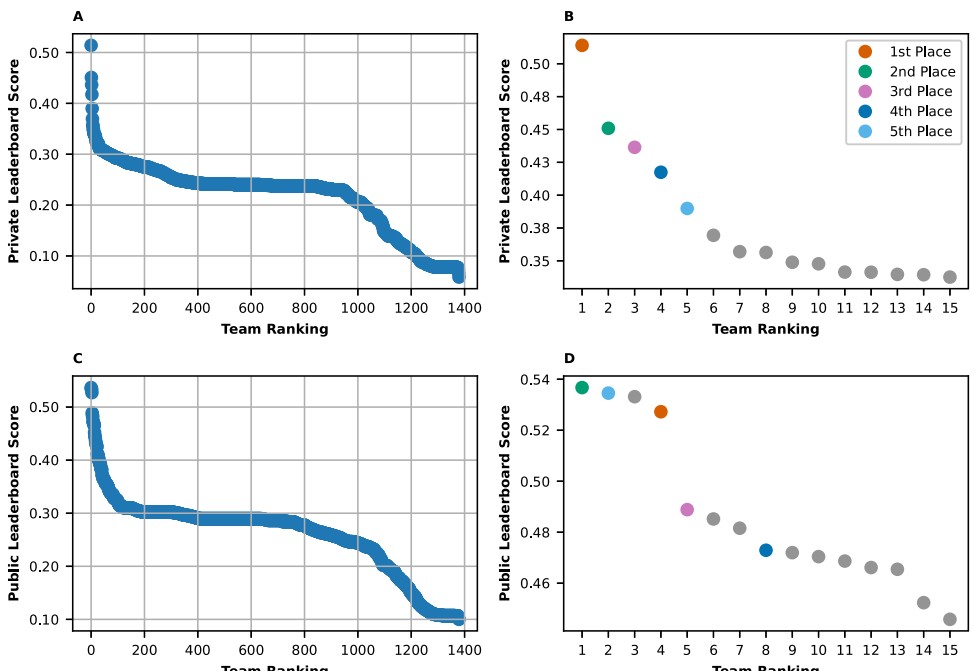

**Fig. 1 | Leaderboard scores as a function of team ranking.** Private and public leaderboard (mean average precision) scores as a function of the performance ranking of the teams. **A** and **B** show the scores on the private test set and **C** and **D** show scores on the public one. **B** and **D** zoom in on the top 15 ranked teams. The top 5 models in the private leaderboard all appeared in the top 8 entries in the public leaderboard, with some variations in the order, as indicated by the color in the zoomed-in figures (right panels). Source data are provided as a Source Data file[71].

total precision largely driven by the Turns class. F1 values revealed a more balanced picture, with all FOG scores ranging between 0.73 and 0.81. As noted above, FOG detection during walking appears to be quite challenging for all models. Start Hesitation performance was generally better.

Table 2 presents the intraclass correlation coefficients (ICC) between model estimates and actual %TF, number of FOG occurrences, and total duration of FOG episodes per subject (average values in each test set and estimated values according to each winning model are shown in Table 3). For %TF, the first three models show excellent[51] validity on the private test set and very good agreement when the public test data is added. All five models show good to excellent results (Table 2), indicating that the models accurately estimated the percentage of the total time spent freezing, and all ICCs were statistically significant ($p < 0.001$). When analyzing the number of freezing episodes, performances on the private test set were good for three of the models (1st, 2nd, and 5th: ICC > 0.75, $p < 0.001$), moderate for the 3rd place model (ICC = 0.72, $p = 0.001$) and poor for the 4th place model (ICC < 0.1 and $p > 0.3$). On the private+public joint test set, performance dropped with ICCs < 0.6 for all models of the number of FOG episodes. Estimating the total FOG duration had the most impressive results, with ICCs > 0.90 ($p < 0.001$) for all models on both the private and the private+public test sets, and ICC > 0.96 on the hidden (private) data.

After the contest, we learned that there was some, unintentional overlap between the training and private test sets. To evaluate performance on a clean dataset without the possible impact of overfitting, the same analyses were conducted while excluding overlapping subjects (five in the private test set and seven in the public test set) between the training and test sets. These results are detailed in the supplementary information (Supplementary Fig. 4 and Supplementary Tables 4–8), including the % of the change in the evaluation and agreement metrics. Briefly, for many models and for key performance measures, the differences were quite small (i.e., less than 5%).

## Daily living, exploratory results

Figure 3A depicts the per hour %TF during walking and periods adjacent to walking bouts estimated by the 1st place model against unlabeled daily living data for 45 people with PD known to have FOG (see the "Methods" section for more details) and 19 people with PD who were not considered to have FOG (see Table 4C in the "Methods" section). A non-parametric Mann–Whitney $U$ test with a Benjamini, Krieger, and Yekutieli correction did not show any statistical differences between the hours of 7:00 to 22:00 (nighttime was excluded from the analysis). To begin to further explore the daily living data and the potential utility of applying the winning models to real-world data, we also created a joint model of the 1st, 3rd, and 5th places (see the "Methods" section for more details). The %TF as a function of time of day estimated by the joint model is depicted in Fig. 3B.

As shown in Fig. 3B, %TF significantly differed in freezers and non-freezers (based on self-report via the NFOG-Q or observation during clinical assessment) at nine hours of the day and two hours at night (using a non-parametric Mann-Whitney $U$ test with a Benjamini, Krieger, and Yekutieli correction). For the freezer group, a Friedman test found that there was a significant difference between the hourly %TF during the daytime hours and the median %TF at night, which was used as a reference for the comparison ($p < 0.001$). A post-hoc analysis (two-stage linear step-up procedure of Benjamini, Krieger, and Yekutieli) identified significant differences in %TF between each daytime hour and the night reference value ($p < 0.02$) after adjusting for multiple comparisons. In both Fig. 3A and B, there appear to be two peaks in the %TF of the freezer group—around 7 a.m. and 10 p.m.

To investigate whether FOG patterns were similar across different days, based on model estimations, ICCs were computed. Daily %TF was consistent across days in both groups: the intraclass correlations were 0.95 (0.92–0.97; $p < 0.01$) for the freezers, and 0.90 (0.82–0.96; $p < 0.01$) for the non-freezers. When comparing the daily %TF between freezers and non-freezers, the effect size of the mean between-group difference was 0.7 (a moderate effect size).

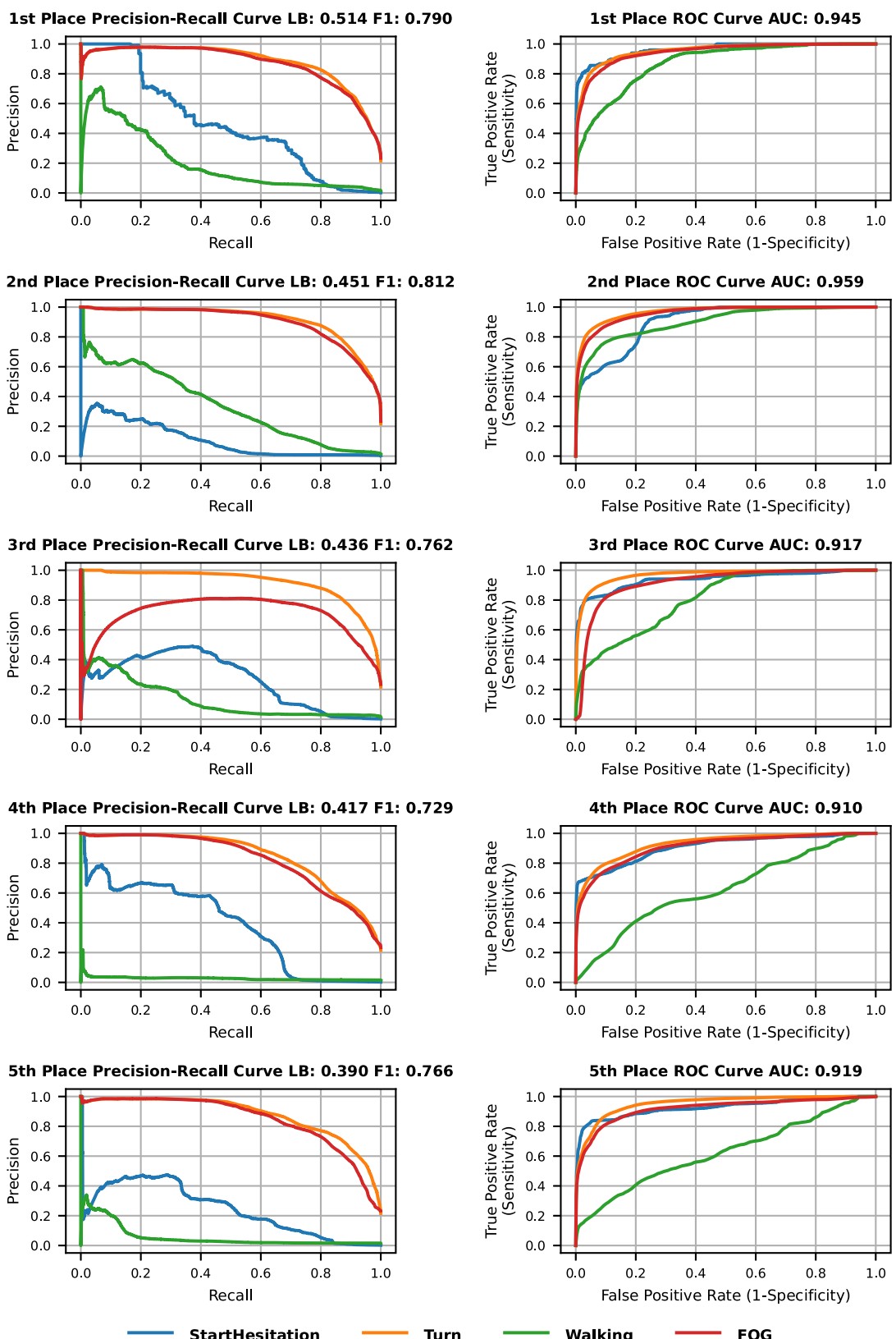

**Fig. 2 | Precision-recall and ROC curves of the winning models on the private test data.** Precision-recall curves (left) and ROC curves (right) for the top five ranked models based on the private (hidden) test set. Curve colors indicate the FOG class, as seen in the figure legend. Source data are provided as a Source Data file[71]. LB leaderboard, ROC receiver operating characteristic, AUC area under the ROC curve.

**Table 1 | Evaluation metrics of all FOG classes of the top 5 models on the private (hidden) test data at the point on the precision-recall curve closest to (1,1)**

| Team | FOG class | F1 score | Accuracy | Precision | Recall | Specificity |
|---|---|---|---|---|---|---|
| 1st place | Start Hesitation | 0.459 | 0.996 | 0.346 | 0.681 | 0.997 |
| | Turn | 0.814 | 0.920 | 0.821 | 0.807 | 0.951 |
| | Walk | 0.079 | 0.682 | 0.041 | 0.915 | 0.678 |
| | All FOG | 0.790 | 0.904 | 0.805 | 0.775 | 0.943 |
| 2nd place | Start Hesitation | 0.014 | 0.674 | 0.007 | 0.960 | 0.673 |
| | Turn | 0.838 | 0.931 | 0.853 | 0.823 | 0.961 |
| | Walk | 0.407 | 0.983 | 0.417 | 0.399 | 0.992 |
| | All FOG | 0.812 | 0.915 | 0.839 | 0.786 | 0.954 |
| 3rd place | Start Hesitation | 0.439 | 0.997 | 0.445 | 0.434 | 0.999 |
| | Turn | 0.843 | 0.933 | 0.855 | 0.832 | 0.961 |
| | Walk | 0.053 | 0.476 | 0.027 | 0.972 | 0.469 |
| | All FOG | 0.762 | 0.885 | 0.738 | 0.788 | 0.915 |
| 4th place | Start Hesitation | 0.495 | 0.998 | 0.583 | 0.430 | 0.999 |
| | Turn | 0.754 | 0.895 | 0.761 | 0.747 | 0.935 |
| | Walk | 0.032 | 0.116 | 0.016 | 0.980 | 0.103 |
| | All FOG | 0.729 | 0.876 | 0.742 | 0.716 | 0.924 |
| 5th place | Start Hesitation | 0.373 | 0.998 | 0.455 | 0.317 | 0.999 |
| | Turn | 0.789 | 0.905 | 0.760 | 0.820 | 0.929 |
| | Walk | 0.031 | 0.071 | 0.016 | 1.000 | 0.057 |
| | All FOG | 0.766 | 0.890 | 0.760 | 0.771 | 0.926 |

Similar results were obtained for these measures when evaluating these models on the private +public test sets (see Supplementary Table 1). As mentioned above, "All FOG" does not refer to the average of the performance in each class, but rather refers to the binary case of FOG vs. non-FOG.

To further explore these daily living results, we also split the group of people with FOG into two groups (severe freezers and moderate freezers) using associated patient-reported data (see Fig. 3C). Comparing the %TF during the day using a Friedman's test showed differences between the three groups ($p < 0.01$). A post-hoc analysis using Dunn's test revealed that all three groups were significantly different from the others (adjusted $p$-value < 0.04).

An additional exploratory analysis compared the real-world distribution of FOG classes detected by the joint model to the distribution in the labeled data based on the video annotations during the FOG-provoking tests (Table 5). Turning FOG was, by far, the most common class in both cases, but it was slightly more common in the daily living data. Start hesitation FOG constituted <1% of the episodes detected in the real-world data and 10% during the FOG-provoking tests. Thus, while turning FOG was the most common and start hesitation FOG was the least common in both datasets, the distributions were similar but not identical.

## Discussion

To improve the automatic detection of FOG during FOG-provoking tests, we embarked on the Parkinson's Freezing of Gait Prediction challenge. The contest attracted over 1300 participating teams from 83 countries, many of whom had no prior experience in the world of PD or FOG. The winning efforts markedly improved previous detection abilities.

On unseen private test data and public test (validation) data, the top winning models showed high agreement (ICC > 0.87) between estimated and actual values of two gold-standard FOG measures (i.e., % TF and total FOG duration). Poorer performance on counting the number of FOG episodes may be due to fragmented detection patterns where continuous events were detected as a series of short events. A post-processing method to merge close events could potentially improve performance. Nonetheless, our findings indicate that even without post-processing, the methods introduced here are successful at identifying FOG events. While the agreement between the winning machine learning model predictions and the raters was not perfect, it was similar to that of previous smaller-scale studies[19,36] and to that obtained when comparing the current gold standard of human raters[52–54]. In other words, from the perspective of inter-rater agreement, these machine-learning methods performed as well as expert raters.

Intraclass correlation measures of reliability can be affected by class imbalance. For a more extensive evaluation, including performance per class, we computed metrics of accuracy, sensitivity, specificity, and precision, as well as F1 and area under the ROC curve. Previous studies that aimed to automatically detect FOG from inertial sensors tend to report sensitivity and specificity (AUC is common as well). When looking at the binary "all FOG" case classifying samples into FOG or non-FOG, accuracy was high, similar to previous studies with a single waist-worn sensor[33,34,41]. The best model in this competition (based on the mean average precision ranking) achieved a sensitivity (recall) and specificity of 0.78 and 0.94, respectively (see Table 1). The 2nd place model scores were slightly higher: 0.79 and 0.95. The 3rd place model was the only one out of the other winners with a higher value in one of the metrics: a sensitivity of 0.79, but specificity slightly dropped to 0.92. Previous studies using similar sensor data have shown somewhat similar performances[33,34,39–42], tending to present a slightly different balance of sensitivity and specificity (here, sensitivity was lower and specificity was higher). The thresholds we used to extract measures from the winning models were selected with a focus on the precision-recall trade-off and the F1-score, while previous FOG detection studies typically aimed to achieve higher sensitivity and specificity values. Also, sample sizes in previous work were smaller[55] and relied mostly on evaluation methods like leave-one-subject-out, which are likely to be overly optimistic and less generalizable as they often rely on total performance using the full dataset when conducting parameter tuning. Thus, from the perspective of these metrics, the winning approaches advance the state-of-the-art.

The present results also move the field forward with regard to the precision achieved in the "all FOG" case. Most previous studies did not report precision or F-scores[56]. Sigcha et al. recently achieved a precision value of 0.62, and an F1-score of 0.71[40]. In another recent study comparing state-of-the-art deep learning approaches, a precision of 0.26 with an F-score of 0.39 was achieved[57]. Here, precision and F1-scores were generally higher than both, ranging from 0.74 to 0.84 for precision and 0.73 to 0.81 for F1-scores. For structured FOG-provoking tests, like those typically conducted in a clinic or research lab, the precision values obtained by the winners of the contest may be sufficient.

Almost all the winning architectures were ensemble-based, combining the power of multiple networks with complementary strengths, to achieve optimal performance and increase generalization. All top five solutions incorporated either gated recurrent unit (GRU) or long short-term memory (LSTM) architectures in their models, to process acceleration sequences in a way that maximizes the ability to capture past contexts, as well as future contexts in the case of the architectures that utilized bidirectional RNN layers (recall the supplementary information). The 1st and 3rd place architectures relied on the abilities of transformers. Based on the concept of self-attention, which models relationships between different parts of a sequence, (e.g., words in a sentence), transformer networks achieved some of the best results so far in FOG detection[40] and hold potential when combined with RNNs.

**Table 2 | Intraclass correlation results reflecting the model's ability to reproduce gold-standard measures based on expert review of the videos**

| ICCs (CI: 95%) | | 1st place | 2nd place | 3rd place | 4th place | 5th place |
|---|---|---|---|---|---|---|
| % Time frozen | Private test | 0.949** (0.85–0.98) | 0.934** (0.80–0.98) | 0.942** (0.83–0.98) | 0.886** (0.69–0.96) | 0.877** (0.67–0.96) |
| | Private+public test | 0.869** (0.77–0.93) | 0.884** (0.79–0.94) | 0.898** (0.82–0.94) | 0.870** (0.77–0.93) | 0.852** (0.74–0.92) |
| No. of FOG episodes | Private test | 0.763** (0.04–0.94) | 0.869** (0.64–0.96) | 0.717** (0.34–0.90) | 0.093 (−0.22 to 0.50) | 0.885** (0.68–0.96) |
| | Private+public test | 0.500** (0.18–0.71) | 0.456** (0.18–0.67) | 0.597** (0.30–0.78) | 0.084 (−0.12 to 0.32) | 0.346* (0.04–0.59) |
| FOG duration | Private test | 0.991** (0.97–1.00) | 0.991** (0.97–1.00) | 0.985** (0.95–0.99) | 0.965** (0.90–0.99) | 0.985** (0.96–1.00) |
| | Private+public test | 0.955** (0.92–0.98) | 0.944** (0.89–0.97) | 0.965** (0.93–0.98) | 0.950** (0.91–0.97) | 0.907** (0.82–0.95) |

*$p < 0.05$, **$p < 0.001$ (exact $p$-values are shown in Supplementary Table 2) in an ICC2(2,1) test. Note that for some models and outcome measures (e.g., 1st place model, no. of FOG episodes), performance when combining the public and private test sets was lower than that seen in the private data. The data was randomly divided into different test sets, so this finding is somewhat counterintuitive. Notably, this occurred for the number of FOG episodes, an outcome that was generally less robust compared to FOG duration or % time frozen (perhaps because the splitting or lumping of adjacent episodes affects the number of episodes much more than the duration of % time frozen).
ICCs intraclass correlation coefficients.

**Table 3 | Measures of FOG occurrences quantified in the two test sets and estimated by the winning models (mean ± SD)**

| Private test: $n = 14$ Private+Public test: $n = 40$ | | % Time frozen | No. of FOG episodes | FOG duration (s) |
|---|---|---|---|---|
| Gold standard, video-based | Private test | 14.39 ± 15.07 | 27.93 ± 22.23 | 184.12 ± 353.49 |
| | Private+public test | 22.64 ± 20.98 | 33.25 ± 25.07 | 273.66 ± 346.89 |
| 1st place | Private test | 12.44 ± 16.88 | 16.36 ± 18.41 | 175.54 ± 374.25 |
| | Private+public test | 23.11 ± 23.38 | 21.00 ± 21.45 | 268.43 ± 351.53 |
| 2nd place | Private test | 11.65 ± 17.98 | 27.71 ± 31.31 | 173.08 ± 371.49 |
| | Private+public test | 19.33 ± 22.16 | 39.87 ± 41.65 | 235.28 ± 337.64 |
| 3rd place | Private test | 13.65 ± 18.05 | 37.21 ± 45.99 | 194.94 ± 401.92 |
| | Private+public test | 24.37 ± 24.72 | 47.80 ± 38.68 | 296.80 ± 391.58 |
| 4th place | Private test | 13.27 ± 17.99 | 130.71 ± 136.82 | 175.42 ± 393.58 |
| | Private+public test | 24.21 ± 23.62 | 164.70 ± 175.38 | 282.60 ± 377.09 |
| 5th place | Private test | 15.67 ± 16.24 | 28.64 ± 26.13 | 185.41 ± 367.78 |
| | Private+public test | 20.28 ± 20.86 | 37.97 ± 37.64 | 224.81 ± 320.78 |

Median values and interquartile ranges (IQRs) can be found in Supplementary Table 3.

However, they require longer run times and more powerful processing units. In the present context, when real-time processing was not a consideration, the 1st and 3rd-place solutions highlight the potential of transformers in FOG detection.

When discussing the results with the winning teams, they noted the importance of the input size or the length of the acceleration sequences fed into the model. They suggested using short sequences for training and longer ones during inference (classification of new data), to provide more contextual information. If a longer inference input size (e.g., >2 min) offers an improved classification ability, this suggests that a FOG event follows a subtle preliminary motor alteration that begins earlier compared to the few seconds that have been previously considered as the "pre-FOG" gait[58]. This does not apply, however, to the gait initiation class, where prior information is unavailable. Perhaps, different approaches are needed to better detect this type of FOG. More generally, accuracy, precision, and F1-scores were good for "all FOG" and for the turning class, the most common provoker of FOG in the FOG-provoking datasets (recall Table 5). However, for the walking and start hesitation classes, improvement is needed for applications where there is specific interest in identifying those events. Of note, previous work has often used input windowing (also referred to as a classification approach, typically, with few seconds-long overlapping windows)[26,32,56]. In contrast, the leading models in this competition introduced a segmentation method that utilizes longer sequences with smaller patch sizes, thus encapsulating more comprehensive contextual information and dependencies within the data.

Shifting the standard approach to methods of this kind apparently improves FOG detection model performance.

Our analyses suggest that the models developed could replace or augment visual, time-consuming video annotations by expert reviewers. The model output could be used independently, as a kind of second or third reviewer, or to flag suspicious annotations. Further, the increased precision, along with the high accuracy of the winning models, as well as the results on daily living data, suggest feasibility in characterizing real-world FOG using a single, inertial wearable unit, although this needs confirmation against ground-truth data. Some previous studies used multiple sensors, multiple locations, and/or multiple sensor modalities like plantar pressure[59,60], electromyography[61], electroencephalogram[62,63], and heart rate variability[64]. Employing these methods could offer further improvements in future detection technologies, at the expense of introducing added complexity, both in the form of physical requirements and further signal processing. These drawbacks are something to consider for real-world applications, due to potentially reduced patient compliance and sensor synchronizing issues. When considering real-time interventions for FOG, detection needs to be computationally efficient and practical. Therefore, single-sensor detection methods need to be scaled and evaluated concerning their future 'context of use'.

Another consideration in this competition is dataset imbalance. It is typical for FOG datasets to feature a small proportion of FOG events compared to the rest of the data. In the real world, preliminary findings

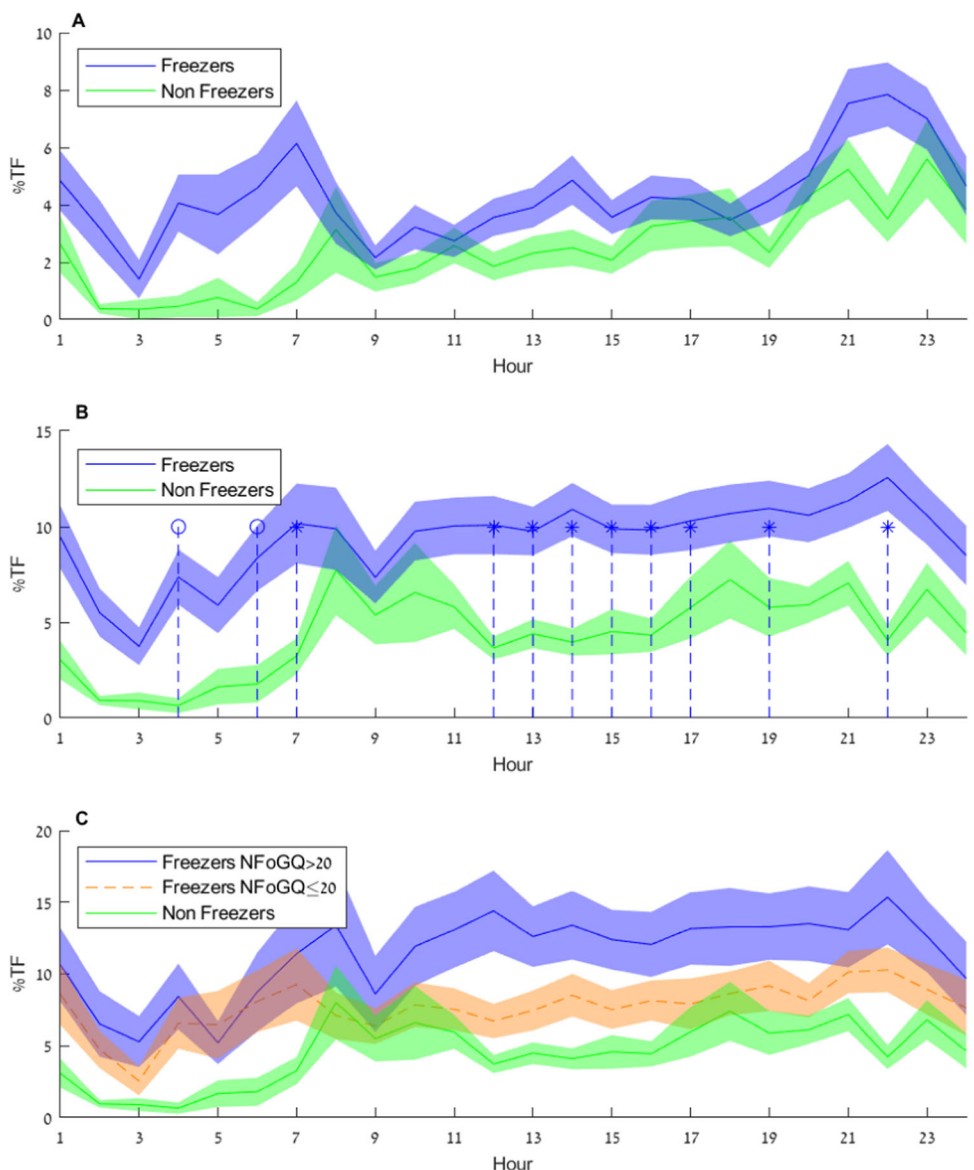

**Fig. 3 | % Time frozen as a function of time of day model estimation on daily living data from freezers and non-freezers.** % Time frozen as a function of time of day model estimation (averaged over the week-long recording) when applied to daily living accelerometer data during and adjacent to walking bouts (see also the "Methods" section). Darker blue, green, and dashed orange lines show the group means, and the shaded areas show the SD. Data in **A** are grouped into people with PD and FOG (freezers, $n = 45$) and people with PD without FOG (non-freezers, $n = 19$), and the values are estimated by the 1st place model. In **B**, data are grouped into freezers and non-freezers, and the values are estimated by a joint model combining the output of the 1st, 3rd, and 5th place winning models, leveraging the strengths of these different approaches (further details and rationale for using this ensemble model are described in the supplementary information). * and ° indicate significant differences between freezers and non-freezers at daytime and nighttime hours (perhaps reflecting FOG that occurs when people get up to use the bathroom during the night[16]), respectively. The groups were compared using a two-tailed non-parametric Mann–Whitney U-test with Benjamini, Krieger, and Yekutieli corrections for multiple comparisons (adjusted $p = 0.087, 0.552, 0.123, 0.025, 0.123, 0.025, 0.037, 0.596, 0.107, 0.058, 0.143, 0.020, 0.018, 0.006, 0.014, 0.014, 0.030, 0.098, 0.025, 0.112, 0.143, 0.006, 0.258, 0.552$ for hours 1 a.m.–12 a.m., respectively).
**C** shows the % time frozen estimated from the output of the joint model, for the severe freezer ($n = 20$), the moderate freezer ($n = 25$), and the non-freezer ($n = 19$) groups. Please keep in mind that ground-truth labels were not available for these daily-living analyses; these results should, therefore, be considered as preliminary findings. Additional statistical comparisons are discussed in the text. Source data are provided as a Source Data file[71]. NFOG-Q New Freezing of Gait Questionnaire.

suggest that patients spend between <1% to 20% of their time in FOG episodes[17–19] (see Fig. 3). The imbalance is partially handled by implementing FOG-evoking protocols, although these protocols can only increase the frequency of events by a limited degree, and they do not solve the issue in real-world, unsupervised experiments. In addition to that, Turn class FOGs are, as previously stated, more common (at least in FOG-provoking testing and recall Table 5). Performance in this class was significantly better than in FOG occurring during gait initiation or walking, despite the usage of a ranking metric that aims to reduce bias

resulting from class imbalance. As most previous work did not explore classification by FOG type, and as turning constitutes the best trigger of FOG in a supervised setting, the present results hold considerable value.

This raises the question of whether the endeavor to investigate FOG by class was helpful. We initially considered introducing the binary classification problem (FOG vs. non-FOG) as the competition's objective. A binary problem might lead to improved performance. The decision to explore FOG class was made with the intention to seek out

**Table 4 | Demographics and characteristics**

**A: Demographics and characteristics of all PD subjects in the private and public test sets**

| Test set | Private (n = 14) | Public (n = 26) | p-value | Test statistics |
|---|---|---|---|---|
| Age (years) | 68.0 ± 7.3 [51–79] | 71.0 ± 8.1 [57–86] | 0.29 | t = −1.158 df = 38 |
| Sex (% F) | 14.3 | 23.1 | 0.21 | χ2 = 1.600, df = 1 |
| Disease duration (years) | 10.3 ± 6.0 [0–20] | 8.7 ± 5.2 [0–19] | 0.26 | t = 0.870, df = 38 |
| MDS-UPDRS Part III on medication | 35.5 ± 7.9 [23–52] | 39.9 ± 15.2 [18–83] | 0.47 | Mann–Whitney U = 187.000 |
| MDS-UPDRS Part III off medication | 47.4 ± 16.0 [30–91] | 44.0 ± 16.1 [12–90] | 0.97 | Mann–Whitney U = 127.500 |
| NFOG-Q | 19.9 ± 3.7 [13–28] | 21.1 ± 3.4 [11–28] | 0.88 | t = −1.093, df = 38 |

**B: Demographics and characteristics of all PD subjects in the FOG-provoking analyses, divided by their original study**

| | tDCS (n = 71) | DeFOG (n = 57) | Combined (n = 128) |
|---|---|---|---|
| Age (years) | 69.9 ± 7.8 [51–94] | 67.9 ± 7.5 [51–86] | 69.0 ± 7.7 [51–94] |
| Sex (% F) | 19.7 | 31.6 | 25.2 |
| Disease duration (years) | 9.2 ± 5.7 [0–23] | 12.0 ± 6.3 [1.5–30] | 10.5 ± 6.1 [0–30] |
| MDS-UPDRS Part III ON medication | 37.6 ± 14.5 [15–83] | 35.0 ± 10.2 [13–56] | 36.1 ± 12.4 [13–83] |
| MDS-UPDRS Part III OFF medication | 43.1 ± 16.9 [12–91] | 43.8 ± 11.7 [18–76] | 43.5 ± 14.3 [12–91] |
| NFOG-Q | 19.4 ± 4.3 [6–29] | 20.4 ± 3.9 [11–28] | 19.8 ± 4.2 [6–29] |

**C: Demographics and characteristics of freezer and non-freezer PD subjects with daily living, real-world data**

| | Freezers (n = 45) | Non-freezers (n = 19) | p-value | Test statistics |
|---|---|---|---|---|
| Age (years) | 67.0 ± 7.0 [51–82] | 66.5 ± 8.0 [55–83] | 0.39 | t = 0.232, df = 63 |
| Sex (% F) | 35.6 | 66.5 | 0.47 | χ² = 0.522, df = 1 |
| Disease duration (years) | 12.4 ± 6.5 [1.5–30] | 5.6 ± 5.2 [0.4–23] | 0.10 | t = 4.104, df = 63 |
| MDS-UPDRS Part III ON medication | 32.7 ± 9.8 [13–56] | 26.7 ± 17.0 [5–72] | 0.09 | t = 1.788, df = 63 |
| MDS-UPDRS Part III OFF medication | 42.4–12.2 [18–76] | – | – | – |
| NFOG-Q | 20.1 ± 4.3 [10–28] | 0 | – | – |

Comparisons were performed using two-tailed Student's t-tests, Mann–Whitney U tests, and Pearson's chi-square tests as appropriate.
*MDS-UPDRS* Movement Disorders Society-Unified Parkinson's Disease Rating Scale, *NFOG-Q* New FOG Questionnaire.

more information, considering the elusive, diverse nature of FOG. The information available in the extant literature about the inherent differences between the FOG classes in acceleration data is insufficient, and we were interested in characterizing the events separately, with the goals of possibly improving model performance and exploring potential clinical ramifications. In fact, however, model performance on the smaller classes was not ideal. Perhaps this was due to the number of events, and even larger datasets with more examples of the other classes could improve the performance of these classes. Nonetheless, the present results do, intriguingly, support the idea that the pattern of the acceleration signals during FOG differs in the turn class, compared to the other FOG classes.

The preliminary analysis of the 7-day daily-living data shows that a joint model combining the output of the 1st, 3rd, and 5th models detects a significant difference in the clinical measure of %TF when comparing PD patients with and without FOG. This difference is apparent during the majority of the daytime (9 out of 16 h). To the best of our knowledge, this is the first time that an analysis of %TF was reported on an hourly basis

**Table 5 | %TF per each FOG class (mean ± SD) during FOG-provoking and daily-living assessments**

| | Start hesitation | Turn | Walk |
|---|---|---|---|
| FOG-provoking tests | 0.7 ± 4.3 | 81.7 ± 28.7 | 17.6 ± 27.6 |
| Daily-living | <0.01 ± < 0.01 | 94.7 ± 5.0 | 5.3 ± 5.0 |
| p-value (comparing FOG-provoking test to Daily-living) | 0.051 | 0.002 | 0.002 |

p-values are based on non-parametric, two related samples, 2-tailed Wilcoxon signed ranks tests. The values are not adjusted for multiple comparisons.

rather than as a daily aggregation[17–19]. Quantifying %TF hourly can provide additional information about on/off motor states due to medication effects (see, for example, ref. [65]), and their impact on FOG throughout the day. Such information cannot be obtained via a snapshot test in the clinic or lab[29]. As suggested by Fig. 3, when most subjects likely wake up in the off-medication state early morning, there appears to be a strong peak in the %TF. In the future, it would be interesting to relate hour-to-hour changes to the time of medication intake, motor response fluctuations[66], and other time varying factors that may contribute to FOG. The models appear to also detect some peaks of %TF among non-freezers during the day.

Additional indirect validation of this combined model was seen when splitting the group of PD patients who suffer from FOG based on self-reported FOG severity (Fig. 3C). The %TF of the more severe freezers was higher compared to those subjects who reported having less FOG. High across-day reliability of daily %TF (ICCs > 0.90) also supports the utility of this joint model.

FOG class distribution was also explored. FOG during turning was detected in 94% of the time spent frozen in a daily living setting (Table 5). This provides the first objective support for the possibility that Turn class FOG is the most common in real-world FOG, similar to its relatively high frequency during FOG-provoking protocols[67]. However, class imbalance in the training dataset may have played a role. It is likely that some Walking and Start Hesitation events were detected as Turn events. Still, the preliminary result that over 90% of predicted FOG occurred during turns suggests, intriguingly, that, even during daily living, turns are the most common trigger of FOG.

Given the many unknowns about the occurrence of FOG, these initial, exploratory findings suggest that reliable measures of real-world FOG can be obtained for which the winning solutions can be the basis. Although there was some overlap of the subjects with freezing in

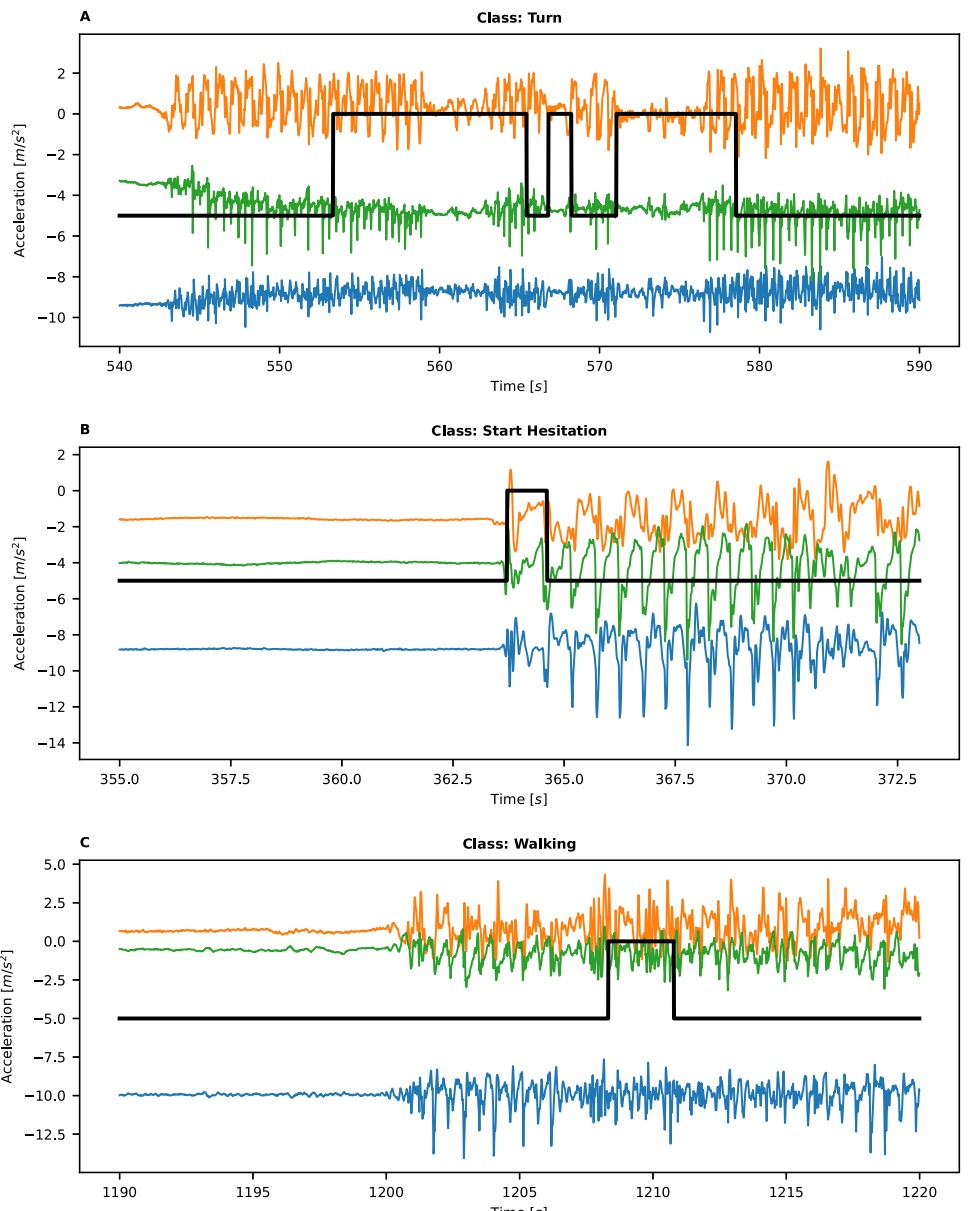

**Fig. 4 | Example of acceleration signals with FOG labels.** Example of 3D acceleration signals with FOG instances and gold-standard labels of FOG (thick black lines) as determined by the review of the videos by expert reviewers. The three accelerometer signals are shown in orange, green, and blue. Occurrences of FOG, as determined by the expert reviewers, are shown by the black line (values are arbitrary). **A** shows a Turning FOG, **B** shows a gait initiation (Start Hesitation) FOG, and **C** shows a FOG event during walking. Source data are provided as a Source Data file[71].

the home-based FOG-provoking test data and the daily living dataset, no labeled training was conducted on real-world data and there was no tuning of any parameters. Nonetheless, improved precision might be required to reduce false detection. Indeed, the models appear to detect some %TF peaks among non-freezers, perhaps because more walking bouts occurred during those hours (8–11 a.m. and 6–8 p.m.), which could increase the false detection of FOG. Thus, additional studies are needed to confirm or reject these preliminary real-world findings and to further develop and validate AI approaches for assessing FOG in daily life, perhaps using ground-truth labels. In the meantime, these preliminary results support the idea that turning FOG is the most common FOG class during daily living and that FOG most likely occurs at specific times of the day and night.

In conclusion, the results of this competition demonstrate the potential of using a single lower-back wearable sensor, in conjunction with AI modeling, to effectively identify and assess FOG across FOG-provoking tests. This approach can be applied widely to reduce the reliance on expert annotations while maintaining high accuracy in FOG detection with minimal trade-offs and without compromising the ability to accurately recognize and quantify FOG. Nonetheless, when there is interest in specific FOG classes, model performance would benefit from improvements. Leveraging the winning methods and insights presented here, further refinement and application-specific tuning can lead to the development of even more robust and precise models and tools. These advancements open the door to the possibility of implementing such solutions in real-world scenarios. Initial, preliminary exploration of that possibility shows encouraging and intriguing time-of-day effects that have not been reported previously. Further, the competition data can now be used as a standard way of assessing new methods that come along, instead of smaller and less heterogeneous existing datasets[56,57]. The tremendous participation of the machine-learning community in contributing to these models

**Table 6 | Class event frequency and duration in different parts of the dataset based on the expert reviews of videos**

| | Train | Public test | Private test | Total |
|---|---|---|---|---|
| Start Hesitation | 107 (39.8)[a] | 53 (3.7) | 16 (0.4) | 176 (43.9) |
| Turn | 1948 (316.4) | 630 (113.2) | 309 (39.5) | 2,887 (469.1) |
| Walk | 395 (43.5) | 262 (22.5) | 66 (3.1) | 723 (69.1) |
| Notype | 1032 (83.2) | – | – | 1032 (83.2) |
| Total | 3482 (482.9) | 945 (139.4) | 391 (43) | 4818 (665.3) |

[a]Entries are the number of FOG events (time [min]).

demonstrates the potential for additional competitions to rapidly develop new solutions to complex medical challenges like FOG, without the need for prior domain knowledge.

## Methods

The competition data were previously collected for other research purposes at three sites and were de-identified for the contest. The original studies received human studies approvals from the local human studies committee at Hebrew SeniorLife (IRB-2016-13), the Tel Aviv Sourasky Medical Center (0710-15-TLV, 0674-15-TLV, 0908-18-TLV), and UZ/KU Leuven (internal reference number: s62453). The reuse of the anonymized data was also approved by the relevant human studies committee. All participants gave their written informed consent before participating in the original data collection studies. Since research to date has not reported any association between the behavioral manifestation of FOG and sex or gender, no sex or gender analysis was carried out. Participant sex is reported based on self-report.

### Competition data

Two labeled datasets (tDCS FOG[68] and DeFOG[16]) and one non-labeled dataset (Daily Living) (see the subsection "Datasets" and Table 4B, C for more information) were made available during the competition. Each of those comprised 3D accelerometer data from a lower back sensor. The labeled data were annotated by expert reviewers based on video recordings[16]. Labels were provided (FOG: yes or no and the FOG class. As FOG is diverse in nature and its occurrence depends on the context, the trigger (or class) of FOG can be informative as well). The three classes of FOG episodes were 'Walking', (FOG episode that has occurred during normal walking), 'Turn', (FOG episode at a turning phase), and 'Start-Hesitation' (FOG episode during gait initiation). In cases where the class was not available, it was defined as 'notype' data. Test data were randomly assigned and split into a public dataset and a private, hidden one (see Table 4A for subject characteristics). The public test set was only public in the sense that performances on it were visible to contestants during the competition period; the data itself was not revealed. The private test set was used for the final evaluation of the models. Representative examples of acceleration signals that contain FOG from each class according to expert review are shown in Fig. 4. Demographic, PD, and FOG characteristics of the labeled data by the original study are shown in Table 4B. Additional information about the dataset can be found on the competition webpage[69].

The total recording time in the combined labeled dataset from tDCS FOG and DeFOG was around 92 h, out of which roughly 64 h were annotated by two trained experts based on videos. The data contained 4818 examples of FOG with a total duration of 665.3 min (over 10 h) (Table 6).

### Datasets

**tDCS FOG**. The tDCS FOG data were originally collected as part of a sham-controlled, double-blinded multi-site randomized trial (Clinical-Trials.org: NCT02656316) examining the effects of transcranial direct

current stimulation on FOG[68]. Included in the competition dataset were 71 patients with PD with mild-to-moderate symptoms (Hoehn and Yahr score of 1-3.5[70]) and self-reported (via the NFOG-Q) FOG, who performed videotaped FOG-provoking tests[15] while wearing a sensor that included a 3D accelerometer placed on the lower back (Opal, APDM Wearable Technologies, Portland, OR, USA. Sampling rate 128 Hz. Acceleration units m/s²). Patients were excluded from the original study if they were unable to ambulate 20 m unassisted, suffered from medical conditions detailed in the original work[68], or were unfit for the treatments. The FOG-provoking test protocol was conducted in the on-medication state and, if the patient agreed, also in the off-medication state. From 71 tDCS FOG patients with three FOG class labels, 833 data series from 62 patients were included in the training data, 138 data series from 14 patients in the public test set, and 68 data series from 7 patients in the private test set (note that the sum of these numbers is larger than 71 due to the inadvertent overlap; see Results and the online data repository[71] for more information). The experts who reviewed the tDCS videos and provided annotations also included a kinetic/akinetic FOG label, which was not relevant to the scoring of the competition. The percentage of akinetic episodes out of the total episodes in this dataset was 5.6%.

**DeFOG**. The DeFOG study was a randomized single-blind, multi-center, controlled trial (ClinicalTrials.gov: NCT03978507) designed to test the effectiveness of a smartphone-based, on-demand cueing-type intervention for FOG[16]. The de-identified data of 57 PD patients with self-reported (via the Characterizing Freezing of Gait Questionnaire (C-FOG)[72]) FOG enrolled in the study were utilized in this competition. Patients with PD and FOG were only included in the original study if they were able to walk for 5 min unassisted by another person, and their modified Hoehn and Yahr stage ranged between 1 and 4 in the on-medication state. See Zoetewei et al.[16] for further details. The recordings were made with a small, lightweight IMU (Axivity Ltd., York, UK; either AX3 or AX6 model, size: ~23.0 × 32.5 × 7.6 mm; weight: ~11 g; 100 Hz sampling rate), while patients performed a FOG-provoking protocol as part of a home-based assessment, both in the off-medication and on-medication state. Annotations were determined based on pre-set definitions and steps were taken (such as several fidelity checks and sharing of scoring) to ensure cross-site consistency[16]. 91 data series with three class labels (each has more than one trial) from 38 patients were included in the training data, 30 data series from 12 patients in the public test set, and 16 data series from 7 patients in the private test set. The training data also contained 46 'notype' data series from 25 patients. For this dataset as well, kinetic/akinetic FOG labels were provided in the competition data, with 29.3% of the episodes being marked as akinetic. Overall, taking into account the DeFOG and tDCS datasets, 18.6% of the episodes were labeled by the experts as akinetic.

**Daily living data**. Data in the daily living dataset came from both the DeFOG study and another study (ONPAR) aimed to investigate prefrontal cortex activation during obstacle negotiation in people with PD who did not have FOG[73]. As part of the DeFOG study protocol, participants were asked to wear a sensor for seven consecutive days to monitor their daily activity without receiving any cueing or feedback from the DeFOG system. A small body-fixed, waterproof sensor that contains a 3D accelerometer (Axivity Ltd., York, UK; either AX3 or AX6 model, size: ~23.0 × 32.5 × 7.6 mm; weight: ~11 g; 100 Hz sampling rate) was secured to their lower back (lumbar vertebrae 4–5) with medical-grade tape for 7 days. These seven days immediately followed the FOG-provoking assessments, which were annotated for the training and testing datasets. We had 45 daily living recordings from 45 of the patients available to include in the dataset. From the latter cross-sectional study (ONPAR), the dataset included 20 recordings from 20 PD patients without FOG who wore an Axivity sensor for 7 days, similar to the DeFOG subjects (one

subject was not included in the post-competition analyses due to a corrupted data file). The ONPAR patients with PD were selected so that age, gender, and disease duration were similar to that of the 45 subjects whose daily living data were available and used in the DeFOG dataset.

## Competition scoring

Submissions included per-sample (time-point) predictions of confidence scores for each of the three FOG classes (turn, walk, start hesitation). Average precision was computed per class and then averaged over the classes to produce the mean average precision, which was the competition evaluation metric. In the case of DeFOG data, only samples that were labeled both Valid and Task (see details on the competition webpage) were included in the evaluation.

## Post-competition analyses of the test sets

We computed the precision-recall curves and ROC curves (along with their AUC) of each of the winning submissions (based on private test set predictions). The analysis was performed per sample in the sequence. F1 score, accuracy, precision, recall, and specificity were computed based on a submission-specific confidence score threshold. The threshold choice aimed to minimize the Euclidean distance between the corresponding point on the precision-recall curve and (1,1), which is the optimal point representing perfect precision and recall. Additionally, we produced percent time frozen (%TF) values, which quantified the percentage of the total time each subject spent in FOG episodes out of the total time included in the submission, an emerging gold-standard[17,19,36]. The number of freezing episodes and their total duration were also extracted. These outcomes were used for evaluating ICC between the estimated and actual %TF, number of FOG, and total FOG duration.

## Daily living analysis

After the competition ended, the winners in the top 5 places were asked to run their models on daily living (24/7) real-world data in order to perform a preliminary analysis of such data. With the exception of the 2nd place group, the rest of the winners provided these results. The analysis included predictions during and adjacent to (five seconds prior to and five seconds after) walking bouts, relying on the assumption that freezing of gait can only occur during those segments. Walking bouts were detected by an algorithm described by Micó-Amigo et al.[74]. The 1st, 3rd, and 5th place models were also used for generating a merged model that detects a FOG event if at least one of the models detected the occurrence of FOG (the 4th model was not included due to its apparent tendency to detect longer FOG episodes as multiple shorter episodes, as evident by its low ICC values for the number of FOG episodes in Table 2 and the high number of FOG episodes values for this model, compared to the others shown in Table 3). Hourly %TF was calculated as the total freezing time divided by the total walking bout time in that hour. Daily %TF was calculated as the total freezing time on that day, divided by the total walking bout time. To compare freezers and non-freezers, the subjects were grouped based on the NFOG-Q or observed FOG during a clinical assessment.

We also tested the prediction of %TF when splitting the freezer group using the NFOG-Q questionnaire score. The median NFOG-Q score was used as the cutoff score (NFOG-Q = 20). The first group, with relatively moderate freezing (NFOG-Q ≤ 20), included 25 subjects (median NFOG-Q = 18 [10–20]), and the second group, of more severe freezers (NFOG-Q > 20), included 20 subjects (median NFOG-Q = 24 [21–28]).

## Statistical analysis

Subject characteristics, including age, sex, disease duration, MDS-UPDRS Part III on and off medication, and NFOG-Q were compared using two-tailed Student's $t$-tests, Mann–Whitney $U$ tests, and Pearson's chi-square tests as appropriate (data were tested for normal distribution using Shapiro–Wilk tests). The tests were performed in SPSS 29.0.0.0.

Intraclass correlation coefficients (ICCs) were computed using the Pingouin 0.5.3 statistical Python package to assess the agreement between the estimated and actual %TF, no. of FOG, and total FOG duration on the private and on the private and public test data. For the daily living analyses, comparisons of the hourly %TF of freezers and non-freezers were performed using a two-tailed non-parametric Mann–Whitney $U$ test with Benjamini, Krieger, and Yekutieli correction for multiple comparisons, due to the non-normal distribution of the hourly %TF. The test was performed in GraphPad Prism 10.0.2. ICC($A,k$) estimates were calculated based on a mean-rating ($k = 6$), absolute-agreement, 2-way mixed-effects model. They were calculated for the total daily %TF to test the stability of this measure across the week (i.e., how similar is %TF from one day to the next). Since not all subjects had 7 full recording days, the ICC was calculated over the first 6 days. Comparisons between the distribution of the FOG classes in the real-world and the FOG-provoking tests were performed using a non-parametric, two related samples, 2-tailed Wilcoxon signed ranks test.

The Python version used in the analyses was Python 3.9. Other packages used for analysis and plotting included Numpy 1.21.5, Pandas 1.5.3, and Scikit-learn 1.2.2. MATLAB R2021b was also used.

## Reporting summary

Further information on research design is available in the Nature Portfolio Reporting Summary linked to this article.

## Data availability

The competition data used in this work are available at the competition webpage: https://www.kaggle.com/competitions/tlvmc-parkinsons-freezing-gait-prediction/data. A copy that includes the test data is available at an open-access database: https://doi.org/10.5281/zenodo.10959560[71]. The figure data generated in this study are provided in the Source Data files at the same link[71]. All other data supporting the findings described in this manuscript are available in the article in the Supplementary Information, in the open-access database, or from the corresponding author upon request.

## Code availability

The code written by competition winners for model training and inference is available at: https://doi.org/10.5281/zenodo.10959560[71].

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

## Acknowledgements

This work and the machine learning competition were supported in part by grants from the Michael J. Fox Foundation for Parkinson's Research (grant ID: MJFF-023293 received by J.M.H.) and from Kaggle. The authors also wish to thank Addison Howard for valuable administrative assistance and the more than 1300 machine learning teams who participated in the competition for their contribution to PD research and for the valuable assistance that they provided to each other.

## Author contributions

E.G., A.S., and J.M.H. conceived of the study. E.G., A.S., J.M.H., L.K., and R.H. organized the machine learning challenge. R.H. designed and implemented the software for leaderboard updating and automated comparison of the results. A.N., P.G., B.M., and J.M.H. facilitated the collection of the data, the collection of the ground truth video labels, and provided the accelerometer datasets. R.H. reviewed the winning models to ensure that they met the previously established rules. Machine learning methods were designed, implemented, run, and described by the participating team members: B.U., H.T., T.Y., S.G., D.L., J.L., and A.K.S. Post-competition analyses were performed by A.S. and E.G. The manuscript was written by A.S., with input from all authors.

## Competing interests

Research support was provided by the Michael J. Fox Foundation to the Tel Aviv Sourasky Medical Center in the form of a grant to help prepare, run, and support the contest. The Michael J. Fox Foundation and Kaggle donated the prize money to facilitate the competition. David Lander is affiliated with TrueFit.AI. L.C.K. and R.H. helped to design the contest and, like all authors, provided feedback on a complete draft of the manuscript. None of the other authors have stocks or shares in companies that are anticipated to gain or lose from the publication of this manuscript, and they declare no competing interests. As per the contest rules, all of the winning teams have licensed the winning Submission and the source code used to generate the Submission (the code for the winning models) under an Open Source Initiative-approved license.
