## [Peer Review File · Nature Communications]

Reviewers' Comments:

Reviewer #1:

Remarks to the Author:

This paper summarizes the effort and results of an extensive machine learning context to predict Freezing of Gait in provided dataset.

The results and methods are highly promising and show excellent validity on the test set, with the potential to really move the field forward.

I have the following minor comments before recommending this paper for publication:

- Why akinetic episodes were excluded? please provide rationale
- The emphasis is strong on the results of the methods applied during daily life, however, primary emphasis should go to the test data and make sure the application to daily living is exploratory (and perhaps not in the main title?). The accuracy seems to be calculated for the test set and not the exploratory daily life, is that correct?
- Figure 4 should also provide the gold-standard FOG label
- Is it possible to move the supplementary materials of daily living to the methods? otherwise it needs to be added somewhere that the FOG detection was done on or around (please specify what is meant by around) gait bouts. Also provide validation information on how the gait bouts are calculated. It is unclear if FOG in daily life was labeled by expert too. Make sure that is declared somewhere.
- the non-freezers seem to have peak of TF at different times, which is interesting. Can some explanation be added about it?

Reviewer #2:

Remarks to the Author:

The article presents the results of the "Parkinson's Freezing of Gait Prediction" challenge.

This challenge is a clear contribution to the state of the art, making available to the research community the largest public dataset of freezing of gait (FoG), attracting interest in automatic assessment of motor symptoms in Parkinson's disease, and providing advances in automatic recognition of FoG using wearable sensors.

The article is well-written and well-organized, with a clear and comprehensive introduction that provides clear context and motivates the need for a public competition; a comprehensive analysis of the results, with proper assessment of classification performance, analysis of the correlation between clinical raters and digital measures, and new insights into the occurrence of FoG in daily life; finally, a detailed and very useful discussion.

I hope that the comments below will help further enhance the quality and importance of this work.

1. Unlike most works in the literature, the FoG recognition problem has been set up as a four-class classification task (start hesitation, FoG during turning, FoG during walking, non-FoG). It is worth clearly stating the motivation for this choice. Moreover,

- a. What is the importance of distinguishing the FoG context in daily life?
- b. FoG during turning is recognized as the most common manifestation of FoG (> 80% of cases). Also in this study, FoG during turning is the most represented class, while the percentage of start hesitations and FoG during walking are negligible in some cases. Is it worth trying to identify such "rare" events instead of focusing on the most common FoG manifestation?
- c. Do the authors believe that inertial signals are somehow different in the three FoG circumstances? Please discuss.

2. The best solutions used deep learning models for FoG prediction, but this is not the main novelty. Instead, a new segmentation approach has been used, which involves analysing time series using longer time sequences/windows/frames and shorter patches. In contrast, windows of 2-5 seconds are often used in the literature, with varying overlaps. This is very different from the best solutions presented in this challenge. It is worth describing and discussing these new approaches, their significance and potential in improving FoG recognition performance.

3. The mean average accuracy (mAP) among the different FoG classes was chosen as the evaluation score.

a. How was it calculated? Why in Table 1 is the average accuracy in the three FoG classes different from that shown in row 139?

b. The best model obtained an mAP of 0.514, which does not allow a distinction between FoG under different circumstances (turning, walking, gait initiation). Ultimately, the results show that FoG (regardless of circumstance) can be detected with good precision and recall. However, the attempt to further distinguish FoG based on context fails. Why did the authors not use a different binary metric (FoG vs non-FoG), such as the F score? Presumably, this would have pushed performance in the direction of maximizing sensitivity and precision in FoG recognition, which has been shown to be the real and possible goal. Please discuss.

4. The results in Table 1, in terms of sensitivity/recall, specificity, and accuracy, are in line with recent deep learning approaches applied to wearable motion data [10.1016/j.eswa.2023.120541]. However, the present dataset includes the largest number of subjects and represents the most heterogeneous dataset. In addition, the precision (which is rarely reported in related studies) is quite high. It is worth noting that this represents a significant improvement over previous approaches.

5. In Table 2, performance is lower when combining the public and private test sets. One might think that the public test set includes more challenging gait patterns and/or more severe subjects. However, from Table 6A, there is no significant difference in clinical scores between the two samples. Please discuss.

6. Table 3 often reports a standard deviation much larger than the mean. Please consider reporting the results in a different way (e.g., median and interquartile range, mean and interquartile range), possibly providing minimum and maximum values as well.

7. How were freezers and non-freezers defined? Based on questionnaires (FOG-Q, NFOG-Q) and/or previous clinical evaluation?

8. In Table 4, the results in daily life are expected. The models were optimized on the training dataset, where the turning FoG was the most represented class. Therefore, they learned better to recognize this class and less the others. In fact, most of the start hesitations and walking FoG were confused with the turning FoG (see precision values in Table 1). When testing the model on unsupervised data, it is expected that most FoG would be recognized as turning FoG. However, this does not necessarily represent the true distribution, but is likely to be a bias introduced by the data used to train the model. This is partially discussed in lines 442-446 and should be better emphasized by taking the results with caution.

9. Lines 505-508. It is stated that "from 70 tDCS FOG patients, 41 patients were included in the training data, 14 patients in the public test set, and 7 patients in the private test set. What about the remaining 8?"

Reviewer #3:

Remarks to the Author:

This manuscript describes an innovative and inspiring contest that invited entrants to develop machine learning methodologies to detect freezing of gait (FOG). The authors are to be commended for the clever idea to hold this contest.

The manuscript is well written. The methods and results are presented clearly and concisely and offer important new insights into how to measure FOG most effectively.

I have no major concerns, but do have a minor point for clarification. Why were the 1st, 3rd and 5th models combined to produce the data shown in Figure 3B? Was there something about those

three models specifically that drove this decision?

Response to the Comments of the Reviewers Nature Communications manuscript NCOMMS-23-60398

We would like to thank the reviewers for their thoughtful evaluation of our manuscript. We are pleased to see that multiple strengths of the manuscript were noted as was the potential to move the field forward. We have carefully revised the manuscript to address all of the comments and suggestions made by the reviewers. We feel that this process has markedly improved the manuscript and its potential to move the field forward. Below we list all of the comments that were raised and describe the changes that were made in response.

REVIEWER COMMENTS

Reviewer #1

Comment: This paper summarizes the effort and results of an extensive machine learning context to predict Freezing of Gait in provided dataset.

Response: Thank you for the positive feedback. We appreciate it.

Comment: The results and methods are highly promising and show excellent validity on the test set, with the potential to really move the field forward.

Response: Great to see that all of the work that went into this contest is valued.

I have the following minor comments before recommending this paper for publication:

Comment: Why akinetic episodes were excluded? Please provide rationale

Response: We apologize if the descriptions surrounding this question were unclear. Akinetic episodes were included in the dataset that was provided in the competition (they are labeled as kinetic or akinetic in the events.csv data file), however, the scoring was not related to these labels. We now explain more clearly in the Supplementary methods description of the labeled data that the expert annotators did provide a label for each FOG episode whether it was akinetic or kinetic. In addition, we now report what percentage of the FOG episodes were labeled as akinetic episodes.

In cases where the expert annotators could not decide, based on the careful review of the video, if an event was an akinetic freezing episode or a voluntary stop, the event was marked as a non-valid event (0 in the "Valid" column), as described in detail in the competition web page:

<https://www.kaggle.com/competitions/tlvmc-parkinsons-freezing-gait-prediction/data>. This information was not part of the scoring of the current machine learning challenge but could be the basis of future study.

Comment: The emphasis is strong on the results of the methods applied during daily life, however, primary emphasis should go to the test data and make sure the application to daily living is exploratory (and perhaps not in the main title?).

Response: We fully agree that the primary emphasis should be on the labeled test data. Indeed, that was the focus of the contest. To address this point, we have added some clarifications in the

introduction, results, and discussion sections to better emphasize the exploratory nature of the daily living analyses and results. In addition, we shortened the discussion of the daily-living results to reduce the emphasis on these findings. While the daily living application is initial and exploratory, we anticipate that future work will use the present results as an initial starting point, as the field shifts its focus to daily living. We have edited the manuscript to make this perspective more clear.

Comment: The accuracy seems to be calculated for the test set and not the exploratory daily life, is that correct?

Response: Yes, that is correct. Evaluation methods against ground-truth labels were only applied to the test sets, as labeled daily living data is currently unavailable. We have edited the text to make this clearer.

Comment: Figure 4 should also provide the gold-standard FOG label.

Response: We apologize for not making it clearer. Figure 4 does show the gold-standard labels. It was meant to present a visual example of what the labels look like. We edited the figure and the its legend to make this clearer.

Comment: Is it possible to move the supplementary materials of daily living to the methods? otherwise it needs to be added somewhere that the FOG detection was done on or around (please specify what is meant by around) gait bouts. Also provide validation information on how the gait bouts are calculated. It is unclear if FOG in daily life was labeled by expert too. Make sure that is declared somewhere.

Response: Thank you for these suggestions. Unfortunately, due to word count limitations, we cannot move this subsection from the supplementary material to the main Methods section of the main manuscript. We added more detail to the description in the supplementary material, as well as a reference to a previous validation of the method for detecting gait bouts. The daily FOG data is unlabeled, as now mentioned more clearly in Results (*Daily Living Results*) and Methods (*Competition Data*).

Comment: the non-freezers seem to have peak of TF at different times, which is interesting. Can some explanation be added about it?

Response: We agree. It does look like the non-freezers have a peak of %TF several times during the day. Unfortunately, we don't have annotations about the participants' activity during their daily living recordings. In one of our analyses of the data, we saw that the non-freezers have more walking bouts at 8-11 am and 6-8 pm, compared to the rest of the hours in the day. We are not sure what activities they performed, but it might have caused more false detections of FOG that led to higher %TF. We edited the discussion to address this comment (although briefly, due to the word count limitation).

(Remarks on code availability):

I reviewed only a couple of codes and seemed to have all the needed info. I did not install it nor tried on data.

Reviewer #2

Comment: The article presents the results of the “Parkinson's Freezing of Gait Prediction” challenge. This challenge is a clear contribution to the state of the art, making available to the research community the largest public dataset of freezing of gait (FoG), attracting interest in automatic assessment of motor symptoms in Parkinson's disease, and providing advances in automatic recognition of FoG using wearable sensors.

Response: Thanks very much for the positive feedback.

Comment: The article is well-written and well-organized, with a clear and comprehensive introduction that provides clear context and motivates the need for a public competition; a comprehensive analysis of the results, with proper assessment of classification performance, analysis of the correlation between clinical raters and digital measures, and new insights into the occurrence of FoG in daily life; finally, a detailed and very useful discussion.

Response: It is quite gratifying to see these favorable remarks. Thank you.

I hope that the comments below will help further enhance the quality and importance of this work.

Comment: Unlike most works in the literature, the FoG recognition problem has been set up as a four-class classification task (start hesitation, FoG during turning, FoG during walking, non-FoG). It is worth clearly stating the motivation for this choice. Moreover,

- a. What is the importance of distinguishing the FoG context in daily life?
- b. FoG during turning is recognized as the most common manifestation of FoG (> 80% of cases). Also in this study, FoG during turning is the most represented class, while the percentage of start hesitations and FoG during walking are negligible in some cases. Is it worth trying to identify such "rare" events instead of focusing on the most common FoG manifestation?
- c. Do the authors believe that inertial signals are somehow different in the three FoG circumstances? Please discuss.

Response: Thank you for these helpful questions and the opportunity to clarify these issues. We did consider the possibility of focusing on the binary case (as discussed also above). However, due to the diverse nature of FOG and the many unknown aspects of its manifestation, we sought an approach that includes more diverse information and decided to explore the aspect of FOG class (or trigger) as well. Sometimes, more classes can improve machine learning results. Indeed, if the movement patterns and hence the acceleration signals differ in the three classes, multi-class models would likely improve the detection of each class of FOG. In addition, the results of the four-class classification task could, potentially, have clinical implications. Indeed, in the current literature, there is not yet enough information about the specific pattern of each FOG class and whether there is an inherent difference in the signals. That is one of the reasons we were interested in the idea of

detecting each class separately. We modified the relevant paragraphs in the methods and the discussion to more fully explain this choice.

As for daily life: most available information today regarding the occurrence, frequency, and nature of each type of FOG is based on FOG provoking protocols. We do not yet know enough about what happens outside of the lab, under unsupervised conditions, during real-world conditions. It is theoretically possible, for example, that start hesitation or walking events are not as negligible during real-world ambulation and this could hold valuable information. Indeed, emerging literature suggests that many aspects of gait and mobility differ in daily living, as compared to when testing is done in the clinic or lab. To be able to address this issue and answer other interesting questions, an objective, widely applicable method of detection of those real-world, daily living events is needed. For example, if certain types of FOG are more common at specific parts of the day, or are more sensitive to specific types of intervention, that information could be beneficial in future research. Of course, models suited for daily living and validation are required first. We have added references to the Introduction to support this idea and modified the Discussion of the revised manuscript to more fully explain this point.

Comment: The best solutions used deep learning models for FoG prediction, but this is not the main novelty. Instead, a new segmentation approach has been used, which involves analysing time series using longer time sequences/windows/frames and shorter patches. In contrast, windows of 2-5 seconds are often used in the literature, with varying overlaps. This is very different from the best solutions presented in this challenge. It is worth describing and discussing these new approaches, their significance and potential in improving FoG recognition performance.

Response: Thank you. We agree that more emphasis should be put on this novel aspect of the winners' solutions. We have modified the discussion appropriately.

Comment: The mean average accuracy (mAP) among the different FoG classes was chosen as the evaluation score.

a. How was it calculated? Why in Table 1 is the average accuracy in the three FoG classes different from that shown in row 139?

b. The best model obtained an mAP of 0.514, which does not allow a distinction between FoG under different circumstances (turning, walking, gait initiation). Ultimately, the results show that FoG (regardless of circumstance) can be detected with good precision and recall. However, the attempt to further distinguish FoG based on context fails. Why did the authors not use a different binary metric (FoG vs non-FoG), such as the F score? Presumably, this would have pushed performance in the direction of maximizing sensitivity and precision in FoG recognition, which has been shown to be the real and possible goal. Please discuss.

Response: Thanks for raising these questions and giving us the chance to better clarify some details.

a. The metric reported in row 139 is the mean average precision which evaluates the precision of the model while taking into account the precision of each FOG class. The precision metric in the table for "All FOG" does not take into account performance in each class equally, but rather looks at general precision when examining the binary case of FOG vs non-FOG. We added a footnote under Table 1 to clarify these points.

b. As discussed above, there were two ideas behind this choice. On the one hand, from a machine learning perspective, it is possible that this choice might improve model performance. On the other

hand, we were also interested in more than just detecting FOG, in general, and hoped to acquire information on the characterization of each FOG class. It is possible that focusing on the binary case would have improved performance, and there's a potential trade-off in that sense. Evidently, detecting two of the classes proved to be quite challenging, in part perhaps due to class imbalance. As mentioned, we modified and edited the discussion to better explain these choices and their ramifications.

Comment: The results in Table 1, in terms of sensitivity/recall, specificity, and accuracy, are in line with recent deep learning approaches applied to wearable motion data [10.1016/j.eswa.2023.120541]. However, the present dataset includes the largest number of subjects and represents the most heterogeneous dataset. In addition, the precision (which is rarely reported in related studies) is quite high. It is worth noting that this represents a significant improvement over previous approaches.

Response: Thank you for this comment. We added more emphasis on this in the revised discussion and included this recent review to further support it.

Comment: In Table 2, performance is lower when combining the public and private test sets. One might think that the public test set includes more challenging gait patterns and/or more severe subjects. However, from Table 6A, there is no significant difference in clinical scores between the two samples. Please discuss.

Response: This is an interesting observation. It is true that the performances on the private test set alone are slightly better for some models and some metrics. The dataset split was determined randomly, however, it is possible that the public test set contains more challenging data samples. Such challenging patterns might arise even in the absence of significant differences in the clinical scores. Please note that the more prominent differences are observed in the "no. of FOG" measure, which is quite unstable by definition, as we mention in the discussion, due to the possibility of detecting long episodes as a series of shorter events, and vice versa. We now speculate about these points in the footnote to Table 2.

Comment: Table 3 often reports a standard deviation much larger than the mean. Please consider reporting the results in a different way (e.g., median and interquartile range, mean and interquartile range), possibly providing minimum and maximum values as well.

Response: Thank you for this observation and suggestion. We have added a new table based on medians and IQR to the supplementary material to address this issue and provide a fuller picture.

Comment: How were freezers and non-freezers defined? Based on questionnaires (FOG-Q, NFOG-Q) and/or previous clinical evaluation?

Response: In the revised manuscript, information has been added in Results and Supplementary Methods to make this clearer.

Comment: In Table 4, the results in daily life are expected. The models were optimized on the training dataset, where the turning FoG was the most represented class. Therefore, they learned

better to recognize this class and less the others. In fact, most of the start hesitations and walking FoG were confused with the turning FoG (see precision values in Table 1). When testing the model on unsupervised data, it is expected that most FoG would be recognized as turning FoG. However, this does not necessarily represent the true distribution, but is likely to be a bias introduced by the data used to train the model. This is partially discussed in lines 442-446 and should be better emphasized by taking the results with caution.

Response: We fully agree. We edited the discussion to clarify that these findings are exploratory and need to be confirmed and supported by future analysis with ground-truth labels.

Comment: Lines 505-508. It is stated that “from 70 tDCS FOG patients, 41 patients were included in the training data, 14 patients in the public test set, and 7 patients in the private test set. What about the remaining 8?”

Response: Thank you for bringing this typo to our attention. The numbers were corrected and clarified in the revised manuscript.

In addition to fixing this typo, this comment prompted us to recheck other things. As a result, we learned that there was a small leakage issue. Five subjects from the training set were included in the private test set, with samples belonging to those subjects constituting 18.5% of the private test set. More precisely, although positive instances of FOG for the same subject were not placed in the training and the private set, there was some overlap in the negative examples due to a misunderstanding about the term “notype”. We have rerun all of the results, after removing this overlap, and now include them in the supplementary material so that the reader can see for his or herself the results with and without the inclusion of this overlap. As we note, for most performance measures, the differences for “All FOG” and for Turns were quite small (less than 5%), suggesting that this mistake had little impact on the results. To further address this issue, we now share the data set on another open-access site (with the link noted in the manuscript), in addition to the Kaggle site. There, we make it clear to people who wish to use the data how to evaluate new models and how to test their models without any overlap between the training and other datasets. They will also be able to compare their results to those of the contest –without any overlap - as those results are now detailed in the supplementary material. We elected to maintain the original results in the manuscript since these reflect the contest results and since the impact was so minimal.

(Remarks on code availability):

- The code for the winning solutions has been made available through public repositories.

Comment: Further comments on the general approach and the choice of some processing procedures (selection of the sequence length and patch size) would be very helpful to the research community.

Response: To address this suggestion, we edited the discussion and supplementary material to place a greater emphasis on the novel aspects of the approaches and processing procedures.

Reviewer #3

Comment: This manuscript describes an innovative and inspiring contest that invited entrants to develop machine learning methodologies to detect freezing of gait (FOG). The authors are to be commended for the clever idea to hold this contest.

Response: Thanks very much for this comment.

Comment: The manuscript is well written. The methods and results are presented clearly and concisely and offer important new insights into how to measure FOG most effectively.

Response: We very much appreciate and value this feedback.

Comment: I have no major concerns, but do have a minor point for clarification. Why were the 1st, 3rd and 5th models combined to produce the data shown in Figure 3B? Was there something about those three models specifically that drove this decision?

Response: Thank you for this comment. We added information to the legend of Figure 3 and also now refer to a more complete explanation in the supplementary methods. The thought to combine winning models derived from the idea that the different models handled the data differently and possibly had different strengths and weaknesses and, therefore, that combining them could potentially improve the overall performance. In a sense, this is an ensemble model, an approach that is increasingly used in machine learning. As we mentioned there, the 2nd place team did not provide us with the results on the daily living data so we could not apply their model. The 4th place model was excluded after considering its performances on the test set – specifically its tendency to treat longer FOG events as a series of shorter events, as explained in the supplementary Methods section.

Reviewers' Comments:

Reviewer #1:

Remarks to the Author:

No further comments

Reviewer #2:

Remarks to the Author:

The authors correctly addressed all reviewers' comments, answering each question and discussing some points better.

In addition, the authors were honest in acknowledging an error in the approach to splitting the data and included the new (corrected) results in the supplementary material. However, I am not sure that keeping the results unchanged in the main paper, even if the difference in performance is limited, is a good idea.

Other than that, the paper can be accepted for publication.

Reviewer #3:

Remarks to the Author:

The authors addressed all of my original concerns with this revision.

**Response to the Comments of the Reviewers Nature Communications manuscript
NCOMMS-23-60398**

REVIEWER COMMENTS

Reviewer #1 (Remarks to the Author):

No further comments

Response: Thank you for your feedback and input.

Reviewer #2 (Remarks to the Author):

The authors correctly addressed all reviewers' comments, answering each question and discussing some points better.

In addition, the authors were honest in acknowledging an error in the approach to splitting the data and included the new (corrected) results in the supplementary material. However, I am not sure that keeping the results unchanged in the main paper, even if the difference in performance is limited, is a good idea.

Other than that, the paper can be accepted for publication.

Response: Thank you for your feedback. With regard to your comment, we believe that it is important to show the original results in the main text since it was the "official" test set based on which the winning entries were ranked in the competition. We find value in including both and feel, therefore, that the new results should be included in the supplementary information. Since the other reviewers did not have this concern, we decided to keep those sections in their current arrangement.

Reviewer #3 (Remarks to the Author):

The authors addressed all of my original concerns with this revision.

Response: Thank you for your feedback.